# Antimicrobial Consumption from 2017 to 2021 in East Trinidad and Tobago: A Study in the English-Speaking Caribbean

**DOI:** 10.3390/antibiotics12030466

**Published:** 2023-02-25

**Authors:** Rajeev P. Nagassar, Narin Jalim, Arianne Mitchell, Ashley Harrinanan, Anisa Mohammed, Darren K. Dookeeram, Danini Marin, Lucia Giangreco, Paola Lichtenberger, Gustavo H. Marin

**Affiliations:** 1Department of Microbiology, Sangre Grande Hospital, The Eastern Regional Health Authority, Sangre Grande, Trinidad and Tobago; 2Country Health Administration, Nariva/Mayaro, The Eastern Regional Health Authority, Rio Claro, Trinidad and Tobago; 3Pharmacy Department, Sangre Grande Hospital, The Eastern Regional Health Authority, Sangre Grande, Trinidad and Tobago; 4Country Health Administration, St Andrews/St David, The Eastern Regional Health Authority, Sangre Grande, Trinidad and Tobago; 5Department of Emergency Medicine, Sangre Grande Hospital, Sangre Grande, The Eastern Regional Health Authority, Sangre Grande, Trinidad and Tobago; 6Independent Researcher, Belmopan, Belize; 7Centro Universitario de Farmacología de Argentina (CUFAR), La Plata 1900, Argentina; 8Division of Infectious Disease, University of Miami Miller School of Medicine, Miami, FL 33136, USA

**Keywords:** antibiotic consumption, Trinidad and Tobago, antibiotics, COVID-19, pandemic, ATC classification, AWaRe, antibiotic consumption, DDD per 1000 inhabitants per day, DDD, Caribbean

## Abstract

An antimicrobial consumption (AMC) study was performed in Trinidad and Tobago at the Eastern Regional Health Authority (ERHA). A retrospective, cross-sectional survey was conducted from 1 November 2021 to 30 March 2022. Dosage and package types of amoxicillin, azithromycin, co-amoxiclav, cefuroxime, ciprofloxacin, levofloxacin, moxifloxacin, nitrofurantoin and co-trimoxazole were investigated. Consumption was measured using the World Health Organization’s Antimicrobial Resistance and Consumption Surveillance System methodology version 1.0, as defined daily doses (DDD) per 1000 population per day (DID). They were also analyzed using the ‘Access’, ‘Watch’ and ‘Reserve’ classifications. In the ERHA, AMC ranged from 6.9 DID to 4.6 DID. With regards to intravenous formulations, the ‘Watch’ group displayed increased consumption, from 0.160 DID in 2017 to 0.238 DID in 2019, followed by a subsequent drop in consumption with the onset of the COVID-19 pandemic. Oral co-amoxiclav, oral cefuroxime, oral azithromycin and oral co-trimoxazole were the most highly consumed antibiotics. The hospital started off as the higher consumer of antibiotics, but this changed to the community. The consumption of ‘Watch’ group antibiotics increased from 2017 to 2021, with a drop in consumption of ‘Access’ antibiotics and at the onset of COVID-19. Consumption of oral azithromycin was higher in 2021 than 2020.

## 1. Introduction

Antimicrobial resistance (AMR) is a major global public health concern. The main driver of AMR development is the misuse and overuse of antimicrobial agents [1]. The Global Action Plan (GAP) to combat antimicrobial resistance was developed and adopted in 2015 by the World Health Assembly [1]. In the GAP, the importance of collecting and analyzing data on antimicrobial use as a means of identifying potential overuse, underuse and inappropriate use of antimicrobial medicines is noted. The GAP also serves as a basis for developing interventions to address inappropriate practices [2,3]. 

Information obtained from antimicrobial consumption (AMC) studies is important for establishing trends in dosing and allowing comparisons of antimicrobial use data. This information enables the development of targeted approaches to help control antimicrobial overuse and develop and implement appropriate strategies to improve antimicrobial use. It provides a basis for countries to develop appropriate policies, regulations, and interventions for the optimal use of antimicrobials. Further, tracking antimicrobial consumption may help to compare data over time and may aid in the assessment of the effectiveness of interventions [2,3,4,5,6,7].

The strategy of the World Health Organization (WHO) for measuring AMC includes a standardized methodology for measuring and reporting national AMC data with common metrics and tools. The standardization of AMC data collection and presentation provides for the monitoring of consumption trends over time, within a country and between countries. This method is incorporated into the AMC module of the WHO GAP and Use Surveillance System (GLASS-AMC), and it can be applied by any country [4,5,6].

This monitoring methodology has been undertaken by developed countries including the European Union as well as Iceland and Norway. All countries have data collected related to the import, procurement, distribution, sales or clinical use of antimicrobials that can serve as a basis for stewardship and monitoring programs [6,7]. However, the surveillance in other countries has not been systematic. Lower resourced countries are thought to be the most challenged because of a lack of standardization, which exposes already vulnerable health systems to failures in antimicrobial stewardship [8]. 

Unfortunately, the Caribbean region is one of those regions in which details of AMC remain unknown. The WHO report on Surveillance of Antimicrobial Consumption in the early implementation (2016–2018) contains no data from Caribbean countries [6]. 

Recently, a report on AMC in six countries in Latin American countries was published following the GLASS-AMC methodology [4]. This publication shows not only the data concerning the six countries in Latin America but also the feasibility of performing this type of study in countries in under-resourced areas, contributing to the inclusion of Latin America among the regions of the world that have periodic, regular and quality data [4].

The data from this study, therefore, provided an avenue to assess antimicrobial consumption within the Caribbean setting. They formed the basis for inferences regarding the risk of multidrug resistance and inappropriate antimicrobial use and providing information for decision making, such as policy prioritization. 

The aim of this study was to measure antimicrobial consumption (AMC) in the Eastern Regional Health Authority (EHRA) of Trinidad and Tobago, according to the ‘AWaRe’ classification, in terms of DDDs per 1000 inhabitants per day (DIDs) using the GLASS-AMC methodology [4,5,6,7,8,9]. It served as a starting point for national data collection capacity-building.

## 2. Methods

This was a retrospective, observational, descriptive study of antimicrobial consumption in the Eastern Regional Health Authority (EHRA) of Trinidad and Tobago and a starting point for national surveillance capacity-building. The quantitative component was aligned to the World Health Organization Global Antimicrobial Resistance and Consumption Surveillance System (WHO GLASS) methodology version 1.0 [7]. The retrospective AMC study was carried out over a 5 month period from 1 November 2021 to 30 March 2022. The retrospective data collected corresponded to the period from 1 January 2017 to 31 December 2021. The WHO GLASS AMC methodology had a core set of antimicrobials for systemic use, following the Anatomical Classification (ATC): Antibacterials (J01), Antibiotics for alimentary tract (A07AA) and Nitroimidazole derivatives for protozoal diseases (P01AB). This excluded topical antimicrobials. 

A selection of antibacterials (J01) were utilized by the ERHA in this sentinel study. The aim was to select a set of antibiotics that were common to the community and hospital settings, respectively, selected by convenience. These sets of antibiotics were used for the sentinel phase of implementation and sentinel surveillance in the ERHA.

Antimicrobials were also classified according to the WHO ‘Access’, ‘Watch’, and ‘Reserve’ (AWaRe) classification; this classification of antibiotics into different groups allowed emphasis on the importance of their appropriate use.

### 2.1. Calculations

The denominator included the population of the ERHA catchment area, i.e., 120,000 persons [10,11,12]. This population coverage represented 8.6% of the country’s total population.

Consumption of antibiotics was summarized as aggregated defined daily doses (DDDs). The standard reporting metric for national estimates was DDDs per 1000 inhabitants per day (DID).

The numbers of DDDs were calculated as follows: Number of DDDs = 𝑻𝒐𝒕𝒂𝒍 𝒈𝒓𝒂𝒎𝒔 𝒖𝒔𝒆𝒅/𝑫𝑫𝑫 𝒗𝒂𝒍𝒖𝒆 𝒊𝒏 𝒈𝒓𝒂𝒎𝒔 where the total in grams of the medicine used was determined by summing the amounts of active ingredient across the various formulations (different strengths of tablets, vials of parenteral medications or capsules, syrup formulations) and package sizes. The DDD value was assigned by the Norwegian Institute of Public Health—Department of Drug Statistics, WHO Collaborating Centre. 

For the purposes of this study, we used the dosage and package types of the antibiotics listed in Table 1.

The ‘Reserve’ antibiotics were only used in hospitals. We sought to capture antibiotics used in both settings (community and hospital); hence, only the ‘Access’ and ‘Watch’ classifications were used, as the ‘Reserve’ classification was used only in hospitals and would not have been captured in community use. All antibiotics were classified in the ‘Access’ and ‘Watch’ categories, as shown in Table 1. 

### 2.2. Training

The Centro Universitario de Farmacología de Argentina (CUFAR) collaborated with the focal point and the pharmacists of the ERHA to provide training sessions for systematic data collection and subsequent automated data analysis. Data were entered in a Microsoft Excel automated data collection tool developed by CUFAR to make the process simple and the quality assured. Training was held in November 2021 via virtual meetings consisting of didactic sessions and discussions about the AMC WHO GLASS methodology and how to obtain the required data to fill cells in the Excel spreadsheets. Refinement of knowledge as well as feedback on data gathering were also provided via email. Reports on the data were provided by CUFAR. The success of this training and analysis was shared with other Caribbean countries in workshops and meetings for further roll out.

### 2.3. Data

Data were evaluated using the WHO methodology for a global program on surveillance of AMC using the ATC classification and the DDD as a standard unit of measurement to express the average maintenance dose per day for a drug used for its main indication in adults. DDDs were transformed into DIDs [7]. 

Data were obtained from the records of community or primary care pharmacies within the St Andrews/St David and Nariva/Mayaro Counties. Secondary care or hospital data were collected from the Pharmacy Department records of the Sangre Grande Hospital. See Figure 1 for a map of the regional health authorities in Trinidad and Tobago. The location of the ERHA is also shown in Figure 1. The health centers included: Biche Outreach Centre, Black Rock Outreach Centre, Brothers Road Outreach Centre, Coryal Outreach Centre, Cumuto Outreach Centre, Grande Riviere Outreach Centre, Guayaguayare Outreach Centre, Manzanilla Outreach Centre, Matelot Outreach Centre, Matura Outreach Centre, Mayaro District Health Facility, Rio Claro Health Centre, San Souci Outreach Centre, Sangre Grande Enhanced Health Centre, Toco Health Centre and 14 h Accident and Emergency and the Valencia Outreach Centre. All health centers were equally chosen, and none were omitted. 

The pharmacists in these facilities entered the data in the WHO GLASS AMC Excel template. Additionally, meetings were arranged to review and analyze the data. 

### 2.4. Assumptions

The total national population was estimated at 1,399,000 inhabitants, and the ERHA population coverage was estimated at 120,000 inhabitants. The total population was extrapolated, as an estimate, from the Population and Housing Demographic report of 2011 and compared with other internationally available data [10,11]. The information on the population of the ERHA was obtained from the Ministry of Heath, Trinidad and Tobago’s website [12]. 

### 2.5. Analysis of Data

The data were validated using the WHO GLASS tool, using the Excel data spreadsheets for each year from 2017 to 2021, and were further analyzed by CUFAR. Antibiotic consumption from 2017 to 2021 was analyzed for each antimicrobial independently and by ‘AWaRe’ category [9]. Antibiotics were also differentiated according to route of administration (oral versus intravenous). Adult and pediatric formulations of amoxicillin, co-amoxiclav and cefuroxime were calculated together and combined. The WHO GLASS tool automatically generated the calculations of DIDs. Final reports per year were generated for the period from 2017 to 2021. 

### 2.6. Ethical Approval

This study was approved by the ethics committee of the Eastern Regional Health Authority.

## 3. Results

### 3.1. General Antibiotic Consumption

There was a steady increase in antimicrobial consumption from 2017 to 2020, as seen in Table 2 and Table 3, from 5.699 DDD per 1000 inhabitants per day (DID) to 6.798 DID. From 2020 to 2021, there was a drop in overall antibiotic consumption, from 6.798 DID to 4.611 DID. The hospital was the larger consumer of antibiotics in 2017, at 3.569 DID. This steadily changed, and the community became the larger consumer of antibiotics, at 4.027 DID, in 2020. Oral co-amoxiclav (1.611 DID in 2017 to 2.326 DID in 2020), oral cefuroxime (0.622 DID in 2017 to 1.667 DID in 2019), oral azithromycin (0.278 DID in 2017 to 0.3271 DID in 2021) and oral co-trimoxazole (2.129 DID in 2017 to 1.261 DID in 2020) were the most highly consumed antibiotics. Oral co-trimoxazole consumption decreased over the period, as seen in Table 2 and Table 3. Ciprofloxacin consumption was also high compared to other antibiotics over the period, increasing from 0.2301 DID in 2017 to 0.448 DID in 2020. Moxifloxacin use was mainly confined to the hospital setting (see Table 2 and Table 3). The highest consumption of moxifloxacin was in 2019, at 0.697 DID. Oral azithromycin use was higher in 2021 than in 2020, at 0.327 DID versus 0.271 DID, respectively. Consumption was higher in the community setting.

Consumption of oral ‘Access’ antibiotics such as co-trimoxazole and co-amoxiclav was higher in 2017, but it decreased over the 5 years compared to ‘Watch’ antibiotics, with more ‘Watch’ antibiotics being used (4.103 DID in 2017 versus 2.237 DID in 2021), as seen in Table 4, Table 5, Table 6, Table 7 and Table 8. The oral ‘Watch’ antibiotics were increasingly consumed over the 5 years (1.316 DID day in 2017 versus 2.237 DID in 2021), as seen in Table 4, Table 5, Table 6, Table 7 and Table 8. The consumption of oral and intravenous ‘Watch’ group antibiotics increased proportionally from 2017 to 2021, but decreased in 2020, as seen in Table 7 (1.476 DID in 2017 to 2.318 DID in 2021). The most highly consumed ‘Watch’ antibiotic was cefuroxime (1.667 DID in 2019) followed by azithromycin (0.327 DID in 2021, similar to 2018 consumption levels at 0.3292 DID). The most highly consumed ‘Access’ antibiotic was co-amoxiclav, at 2.326 DDD per 1000 inhabitants per day in 2019. Table 4, Table 5, Table 6, Table 7 and Table 8 show the antimicrobial consumption in Trinidad and Tobago (ERHA) during the years 2017 to 2021, as well as the total data, according to the ‘Aware’ classification. 

### 3.2. Breakdown of Antibiotic Consumption by Year

#### 3.2.1. 2017

For the year 2017, total antimicrobial consumption in Trinidad and Tobago, Eastern Regional Health Authority (ERHA), was 5.699 DID, which allowed the inference that 683 people were treated with some antibiotic daily during this period. Of the 5.699 DID total, 2.169 DID corresponded to the community level, and 3.568 DID to the hospital level. Oral formulations accounted for 95.1% of consumption at the total level (see Table 4). Analyzing the data according to the ‘AWaRe’ classification, at the total level it was observed that 64.52% of the antimicrobials consumed corresponded to the ‘Access’ group, and 35.48% belonged to the ‘Watch’ group (see Table 4).

#### 3.2.2. 2018

For the year 2018, total antimicrobial consumption in Trinidad and Tobago, Eastern Regional Health Authority (ERHA), was 7.097 DID, which allowed the inference that 851 people were treated with some antibiotic daily during this period. Of the 7.097 DID total, 3.376 DID corresponded to the community level, and 3.794 DID to the hospital level. Oral formulations accounted for 95.74% of consumption at the total level (see Table 5). Analyzing the data according to the ‘AWaRe’ classification, at the total level it was observed that 40.23% of the antimicrobials consumed corresponded to the ‘Access’ group, and 59.77% belonged to the ‘Watch’ group (see Table 5).

#### 3.2.3. 2019

For the year 2019, total antimicrobial consumption in Trinidad and Tobago, Eastern Regional Health Authority (ERHA), was 6.186 DID, which allowed the inference that 742 people were treated with some antibiotic daily during this period. Of the 6.186 DID total, 3.245 DID corresponded to the community level, and 2.941 DID to the hospital level. Oral formulations accounted for 93.84% of consumption at the total level (see Table 6). Analyzing the data according to the ‘AWaRe’ classification, at the total level it was observed that 43.42% of the antimicrobials consumed corresponded to the ‘Access’ group, and 56.58% belonged to the ‘Watch’ group (see Table 6).

#### 3.2.4. 2020

For the year 2020, total antimicrobial consumption in Trinidad and Tobago, Eastern Regional Health Authority (ERHA), was 6.798 DID, which allowed the inference that 815 people were treated with some antibiotic daily during this period. Of the 6.798 DID total, 4.027 DID corresponded to the community level (59.24%), and 2.771 DID to the hospital level (40.76%). Oral formulations accounted for 97.41% of consumption at the total level, 98.98% at the community level, and 95.16% at the hospital level (see Table 7). Analyzing the data according to the ‘AWaRe’ classification, at the total level it was observed that 64.52% of the antimicrobials consumed corresponded to the ‘Access’ group, and 35.48% belonged to the ‘Watch’ group (see Table 7).

#### 3.2.5. 2021

For the year 2021, total antimicrobial consumption in Trinidad and Tobago, Eastern Regional Health Authority (ERHA), was 4.611 DID, which allowed us to infer that 553 people were treated with some antibiotic daily during this period. Of the 4.611 DID total, 2.793 DID corresponded to the community level, and 1.818 DID to the hospital level. Oral formulations accounted for 95.86% of consumption at the total level. Analyzing the data according to the ‘AWaRe’ classification, at the total level it was observed that 49.72% of the antimicrobials consumed corresponded to the ‘Access’ group, and 50.27% belonged to the ‘Watch’ group (see Table 8).

## 4. Discussion

It is important to note that this study used data from pharmacy dispensing records, similarly to Bolivia, Canada, Peru, Kazakhstan, Brunei and New Zealand. Thus, the data collection used in this study has been validated in other countries across the world [4,6]. Data collected between 2014 and 2018 in Europe allowed the estimate that the DID varied by country from 8.9 to 34.1. This study found that the Republic of Trinidad and Tobago was at the lower end of the spectrum from 4.6 to 6.9 DID for the period of data collection [13,14,15,16,17,18,19]. Notably, while some of this may be explained by the decreased levels of health service utilization during the COVID-19 pandemic, it remains important that long-term standardized data collection be continued in the region to gauge the evolution of the patterns of use. This is further underscored by the finding that variation existed in some European countries when assessed longitudinally, in the ‘Access’, ‘Watch’ and ‘Reserve’ classifications, which ultimately suggested that successful implementation of any interventions must be long-term [16,18,19]. As was pointed out in the European report, sustained actions to decrease antimicrobial resistance are predicated on educational activities geared towards the decrease in consumption patterns [16]. One such example is an assessment of antibiotic consumption in Europe between 1997 and 2017, which comprised the most comprehensive description of consumption, with significantly high reporting from all members of the European Union [16,18,19]. This augurs well for a similar model of education to control antibiotic consumption, to be applied in the Caribbean and aligned with the Global Action Plan. 

The increased consumption of ‘Watch’ intravenous and oral formulations seen in this study was similar to that in the global findings from 76 countries [13]. In the ERHA, there was an increase in consumption of oral ‘Watch’ antibiotics from 1.316 DID in 2017 to 3.262 DID in 2019. Consumption of these oral ‘Watch’ antibiotics was still higher in 2021, at 2.237 DID. With regards to intravenous formulations, for the ‘Watch’ group there was an increase in consumption from 0.160 DID in 2017 to 0.238 DID in 2019, with a subsequent drop in consumption in 2020 and 2021. Thus, this increase in consumption was noted for both oral and intravenous formulations, with the community being the larger consumer from 2018 onwards. Similarly, to our study, a study from Nepal uncovered an increase in the use of intravenous ‘Watch’ antibiotics over the period from 2017 to 2019 [17]. A study from Kazakhstan for the period from 2017 to 2019 also uncovered an increase in the use of ‘Watch’ and ‘Reserve’ antibiotics [18]. This was similar to findings from larger studies in Europe and Asia, where ‘Watch’ antibiotics accounted for the largest share of the total consumed [13,14,15,16,17].

In 2019, Khan et al. conducted the first study on antibiotic consumption at one health center in Trinidad and Tobago for retrospective data collection spanning 8 years [20]. Our findings had some similarities to the study by Khan et al., where we noted that the top three consumed antibiotics were oral co-amoxiclav, cefuroxime and azithromycin. The highest consumption seen in the study by Khan et al. was “4.603 DDD per 1000 residents per day”, and the lowest was “1.986 DDD per 1000 residents per day.” This study was conducted in the community, and consumption in the community in 2021 was still higher than in 2017, in our study, with the introduction of curtailment of services due to the COVID-19 pandemic. Thus, the consumption noted in our study was similar. This is interesting, as the study by Khan et al. was conducted in one health center in the North Central Regional Heath Authority (NCRHA), one of the five regional health authorities (RHAs) in Trinidad and Tobago [20]. Additionally, our study was more comprehensive, involving all health centers within the ERHA. Supporting our findings were studies from Europe, Africa and Asia, where penicillin and cephalosporin were found to be the highest consumed antibiotics [14,15,16,17,18,19,21].

In general, the results of this study are similar to those of other cross-sectional assessments of consumption patterns, with an overall increase in the DDD values but a decrease from 2020 to 2021. However, particularly concerning was the increased use of ‘Watch’ categories, which should, therefore, prompt recommendations for rational use [22,23,24]. The literature also supports the use of monitoring tools as a means of controlling consumption through the creation of prescribing targets supported by protocols and treatment guidelines [13,14,15,16,17,18,19]. This point aligns well with the aim of health systems to ensure clinical effectiveness. Thus the ‘Watch’ antibiotics should be preserved, and programs should be targeted at this category for decreased consumption and utilization. The utilization of such a standardized system will provide healthcare administrators with the tools to audit their performance accurately and efficiently. For instance, in the 1997–2017 studies mentioned, a sub-analysis demonstrated that the pattern of quinolone and cephalosporin utilization in Europe fluctuated in concordance with respiratory infections and seasonal variations [19,22]. This allowed for targeted intervention. In fact, in Trinidad and Tobago, an increased usage of fluoroquinolones was observed from 2018 to 2020. This was higher in the community setting. Notably, there was a particularly higher usage of moxifloxacin from 2019 to 2020, even prior to the COVID-19 pandemic. This has been observed in Asia and Eastern Europe but not in all countries of the European Union [15,16,17,18]. Fluoroquinolones such as ciprofloxacin and moxifloxacin were found to be highly consumed in the community setting in southern Europe [22,23]. However, in the ERHA in Trinidad and Tobago, the consumption of moxifloxacin was only an issue for the hospital in 2018 and 2019, whereas ciprofloxacin and levofloxacin consumption was an issue for the community. This is important for antibiotic stewardship efforts. Notably, there is a paucity of literature on the consumption of respiratory fluoroquinolones during the COVID-19 pandemic and in Trinidad and Tobago [22,23,24]. This study showed greater consumption of ciprofloxacin in the community and decreased consumption of moxifloxacin in the hospital setting in 2020 and 2021. This could have been because moxifloxacin is reserved by the ERHA for hospital use, as it is a broad-spectrum antibiotic. This would have helped in stewardship efforts to preserve its use for respiratory illnesses and complicated intra-abdominal infections in patients presenting to hospitals. Trinidad and Tobago also has a high prevalence of tuberculosis, and moxifloxacin may be reserved for drug-resistant infections [25,26]. 

A tool such as this, provided by CUFAR, allows for a powerful level of quality assurance and standardization. It promoted strengthened clinical governance in the antimicrobial resistance section of the health system. This will be particularly useful in Trinidad and Tobago, where there is a paucity of data on this topic and the traditional manner of data collection from the public sector potentially grossly underestimates the actual magnitude of the problem [13]. This pattern is evident in most developing countries, and the implementation of the system in this study may serve as a template for the implementation of similar programs in countries with similar challenges and for primary care and even secondary and tertiary care [27,28,29,30,31,32]. 

A study in Croatia uncovered an increased utilization of azithromycin during the COVID-19 pandemic [30]. Our study shared such findings; however, the consumption of azithromycin was high overall during the period from 2017 to 2021 in the ERHA. In fact, there was higher consumption of azithromycin, particularly in the community setting, in 2021 compared to 2020 in the ERHA, possibly corroborating the increased use of this antibiotic during the COVID-19 pandemic. This finding of increased azithromycin use has also been corroborated by other studies in Brazil and internationally [30,31,32]. However, one systematic review, by Ayerbe et al., did not support the use of azithromycin in patients with COVID-19, leaving room for consideration of stewardship activities in Trinidad and Tobago and internationally [31]. The study by Khan et al. supports our finding of high azithromycin consumption in Trinidad and Tobago even before the pandemic [20]. One study reviewing the global impact of the COVID-19 pandemic on areas such as antibiotic consumption/use showed a global increase in antibiotic use; this was not the case in the ERHA and higher-income countries [29]. Trinidad and Tobago are considered a high-income country. This decreased consumption was because of the curtailment of services such as surgical serves and clinics, similar to the findings of the global survey [29]. In our study in the ERHA, we found that co-trimoxazole was a highly utilized oral antibiotic in the community and hospital settings, but this decreased over the period reviewed. Conversely, studies in Europe from 1997 to 2017 uncovered a seasonal variation in antibiotic use, with a variation in use of antibiotics such as co-trimoxazole, cephalosporins and quinolones [19,22]. The Republic of Trinidad and Tobago is situated in the Caribbean and does not have such seasonal variation, which may have accounted, in part, for the differences seen. Consumption of co-trimoxazole was also high in some parts of the Western Pacific Region [28]. 

## 5. Limitations

The antibiotics were conveniently selected and thus do not represent the total consumption pattern for the Eastern Regional Heath Authority (ERHA). Additionally, the selection is not representative of the entire country’s antibiotic consumption. It only represents the consumption pattern for the selected antibiotics of the ERHA according to the WHO methodology for calculating antibiotic consumption. This, however, was the sentinel site for building national and regional AMC capacity.

## 6. Conclusions

This study highlights the usefulness of the DID calculation, as there are useful comparisons with studies performed internationally. It highlights the decreased consumption seen with the onset of the COVID-19 pandemic, also seen in other high-income countries. 

Starting with a few antibiotics makes for robust data collection and refinement of the methodology. This can be further expanded in the future, as the in-country skill set and capacity is built, to include other sites in Trinidad and Tobago and further, encourage the application of this methodology throughout Latin America and the Caribbean.

## Figures and Tables

**Figure 1 antibiotics-12-00466-f001:**
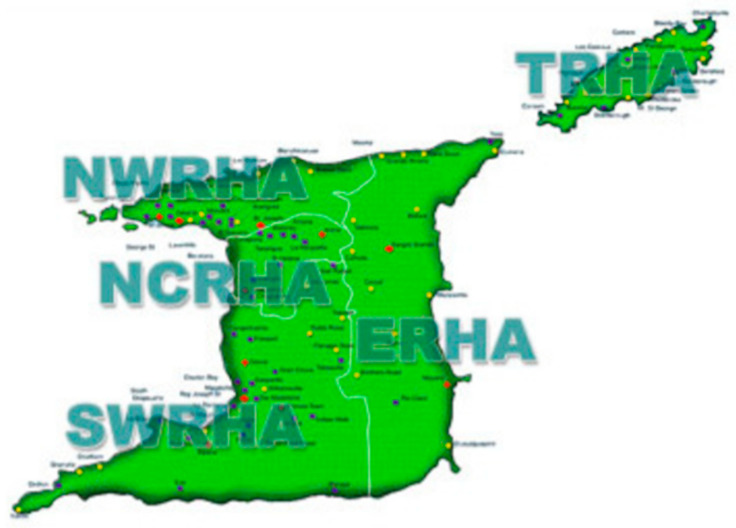
Map of regional health authorities in Trinidad and Tobago.

**Table 1 antibiotics-12-00466-t001:** List of selected antibiotics and ‘AWaRe’ classification.

Antibiotic	‘AWaRe’ Classification [9]
Amoxicillin	‘Access’
2. Azithromycin	‘Watch’
3. Co-Amoxiclav	‘Access’
4. Cefuroxime	‘Watch’
5. Ciprofloxacin	‘Watch’
6. Levofloxacin	‘Watch’
7. Moxifloxacin	‘Watch’
8. Nitrofurantoin	‘Access’
9. Co-trimoxazole	‘Access’

‘AWaRe’ means ‘Access’, ‘Watch’, ‘Reserve’ [9].

**Table 2 antibiotics-12-00466-t002:** Total Antibiotic Consumption (in DID) by the Hospital and Community in 2017 and 2018.

			2017			2018		
ATC5 Code	Active Pharmaceutical Ingredient	ROUTE	Total	Community	Hospital	Total	Community	Hospital
J01CA04	Amoxicillin	O	0.364	0.219	0.145	0.457	0.34	0.117
J01CR02	Co-Amoxiclav	O	1.611	1.029	0.582	1.85	1.204	0.72
J01CR02	Co-Amoxiclav	P	0.12	0.007	0.113	0.120	0.01	0.111
J01DC02	Cefuroxime	O	0.622	0.375	0.286	1.197	0.799	0.398
J01DC02	Cefuroxime	P	0.047	0.0004	0.047	0.016	0.0004	0.015
J01DD04	Ceftriaxone	P	0.052	0.003	0.049	0.120	0.005	0.115
J01EE01	Trimethoprim-Sulfamethoxazole	O	2.129	0.103	2.026	0.429	0.086	0.343
J01FA10	Azithromycin	O	0.278	0.166	0.112	0.329	0.184	0.145
J01MA02	Ciprofloxacin	O	0.231	0.122	0.108	0.772	0.588	0.184
J01MA02	Ciprofloxacin	P	0.025	0.001	0.025	0.013	0.001	0.012
J01MA12	Levofloxacin	O	0.183	0.144	0.039	1.470	0.16	1.311
J01MA12	Levofloxacin	P	0.036	0.0004	0.035	0.008	0.0001	0.008
J01MA14	Moxifloxacin	O	0.002	0	0.002	0.292	0.000	0.292
J01MA14	Moxifloxacin	P	0	0	0	0.0244	0.0000	0.024
	TOTAL		5.699	2.169	3.569	7.097	3.376	3.794

ATC5–WHO ATC code substance level, O—Oral, P—Parenteral. DDD—Defined Daily Dose, DID— per 1000 population per day.

**Table 3 antibiotics-12-00466-t003:** Total Antibiotic Consumption (in DID) by the Hospital and Community from 2019 to 2022.

			2019			2020			2021		
ATC5 Code	Active Pharmaceutical Ingredient	ROUTE	Total	Community	Hospital	Total	Community	Hospital	Total	Community	Hospital
J01CA04	Amoxicillin	O	0.355	0.304	0.051	0.702	0.4907	0.212	0.266	0.207	0.059
J01CR02	Co-Amoxiclav	O	1.886	1.286	0.6	2.326	1.925	0.401	1.649	1.210	0.439
J01CR02	Co-Amoxiclav	P	0.142	0.021	0.121	0.097	0.024	0.072	0.11	0.011	0.099
J01DC02	Cefuroxime	O	1.667	0.986	0.682	1.386	0.909	0.477	1.405	0.824	0.581
J01DC02	Cefuroxime	P	0.053	0.0004	0.053	0.01	0.0009	0.009	0.006	0.0007	0.005
J01DD04	Ceftriaxone	P	0.116	0.006	0.110	0.027	0.015	0.012	0.058	0.005	0.053
J01EE01	Trimethoprim-Sulfamethoxazole	O	0.303	0.074	0.228	1.261	0.12	1.142	0.268	0.103	0.166
J01FA10	Azithromycin	O	0.395	0.228	0.167	0.271	0.222	0.049	0.327	0.191	0.136
J01MA02	Ciprofloxacin	O	0.312	0.158	0.154	0.448	0.319	0.129	0.260	0.129	0.131
J01MA02	Ciprofloxacin	P	0.016	0.00	0.015	0.010	0.001	0.009	0.009	0.0008	0.009
J01MA12	Levofloxacin	O	0.192	0.180	0.011	0	0	0	0.135	0.112	0.022
J01MA12	Levofloxacin	P	0.023	0.0006	0.023	0.0004	0.0004	0	0.0003	0.0003	0
J01MA14	Moxifloxacin	O	0.696	0	0.696	0.228	0	0.228	0.11	0	0.11
J01MA14	Moxifloxacin	P	0.03	0	0.03	0.032	0	0.032	0.008	0	0.008
	TOTAL		6.186	3.245	2.941	6.798	4.027	2.771	4.611	2.793	1.818

ATC5—WHO ATC code substance level, O—oral, P—parenteral, DID—defined daily dose (DDD) per 1000 population per day.

**Table 4 antibiotics-12-00466-t004:** Antimicrobial consumption in Trinidad and Tobago (ERHA) during the year 2017, total data, according to AWaRe classification, expressed in DID (DDD/1000 inhabitants/day).

AWaRe Category	DID According to Route of Administration		Total DID (%)
	Oral	Parenteral	
Access	4.103	0.119	4.223 (74.1%)
Watch	1.316	0.160	1.476 (25.9%)
Reserve	0	0	0
TOTAL	5.420 (95.10%)	0.279 (4.90%)	5.699 (100%)

WHO Access, Watch, Reserve (AWaRe) classifications of antibiotics for evaluation and monitoring of use, 2021. Geneva: World Health Organization; 2021 (WHO/HMP/HPS/EML/2021.04).

**Table 5 antibiotics-12-00466-t005:** Antimicrobial consumption in Trinidad and Tobago (ERHA) during the year 2018, total data, according to AWaRe classification, expressed in DID (DDD/1000 inhabitants/day).

AWaRe Category	DID According to Route of Administration		Total DID (%)
	Oral	Parenteral	
Access	2.735	0.120	2.855 (40.23%)
Watch	4.061	0.181	4.242 (59.77%)
Reserve	0	0	0
TOTAL	6.795 (95.74%)	0.301 (4.24%)	7.097 (100%)

WHO Access, Watch, Reserve (AWaRe) classification of antibiotics for evaluation and monitoring of use, 2021. Geneva: World Health Organization; 2021 (WHO/HMP/HPS/EML/2021.04).

**Table 6 antibiotics-12-00466-t006:** Antimicrobial consumption in Trinidad and Tobago (ERHA) during the year 2019, total data, according to AWaRe classification, expressed in DID (DDD/1000 inhabitants/day).

AWaRe Category	DID According to Route of Administration		Total DID (%)
	Oral	Parenteral	
Access	2.543	0.142	2.686 (43.42%)
Watch	3.262	0.238	3.500 (56.58%)
Reserve	0	0	0
TOTAL	5.805 (93.84%)	0.381 (6.16%)	6.186 (100%)

WHO Access, Watch, Reserve (AWaRe) classification of antibiotics for evaluation and monitoring of use, 2021. Geneva: World Health Organization; 2021 (WHO/HMP/HPS/EML/2021.04).

**Table 7 antibiotics-12-00466-t007:** Antimicrobial consumption in Trinidad and Tobago (ERHA) during the year 2020, total data, according to AWaRe classification, expressed in DID (DDD/1000 inhabitants/day).

AWaRe Category	DID According to Route of Administration		Total DID (%)
	Oral	Parenteral	
Access	4.289	0.097	4.386 (64.52%)
Watch	2.333	0.079	2.412 (35.48%)
Reserve	0	0	0
TOTAL	6.622 (97.41%)	0.176 (2.59%)	6.798 (100%)

The 2019 WHO AWaRe classification of antibiotics for evaluation and monitoring of use. Geneva: World Health Organization; 2019. (WHO/EMP/IAU/2019.11).

**Table 8 antibiotics-12-00466-t008:** Antimicrobial consumption in Trinidad and Tobago (ERHA) during the year 2021, total data, according to AWaRe classification, expressed in DID (DDD/1000 inhabitants/day).

AWaRe Category	DID According to Route of Administration		Total DID (%)
	Oral	Parenteral	
Access	2.183	0.110	2.293 (49.72%)
Watch	2.237	0.081	2.318 (50.27%)
Reserve	0	0	0
TOTAL	4.420 (95.86%)	0.191 (4.14%)	4.611 (100%)

WHO Access, Watch, Reserve (AWaRe) classification of antibiotics for evaluation and monitoring of use, 2021. Geneva: World Health Organization; 2021 (WHO/HMP/HPS/EML/2021.04).

## Data Availability

Data sharing not applicable. No new data were created or analyzed in this study. Data sharing is not applicable to this article.

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
