# Peer review of "Antimicrobial Consumption from 2017 to 2021 in East Trinidad and Tobago: A Study in the English-Speaking Caribbean"

_antibiotics, 2023, doi:10.3390/antibiotics12030466_

Round 1

Reviewer 1 Report

This is an interesting study evaluating antibiotic consumption in East Trinidad & Tobago. Some comments should be addressed as listed below.

Major comments:

1.      Abstract: The abstract is too long as the word count exceeds the journal's recommendation of 200 words. Please revise and shorten it. Also, remove the subheadings as it should be unstructured.

2.      Introduction (line 83): Please cite the GLASS-AMC reference.

3.      Introduction (line 93): Please cite the WHO report.

4.      Methods (line 139): Please cite the website.

5.      Table 1: Please cite the reference of this classification.

6.      Table 2: Please write "in DID" in the table legend. Also, I think it would be more clear to make the table wide as Table 3.

7.      Tables 2 and 3: I think it would be better to report the numbers in three decimals only instead of four (three numbers after the decimal point). This could also help shrink the table and allow space for other columns (i.e., active pharmaceutical ingredient). The same applies to the whole text of the manuscript.

8.      Part 3.1 (Training) is a duplicate. Please remove it from here. However, you may copy the procedure of the training to part 2.2 in the methods.

9.      Results (lines 200-207): This paragraph belongs to the methods section. It should be moved to section 2.3 while avoiding duplicate info.

10.   Line 200: Please justify why the data were not collected from the same period 2017-2021? Or is this the period when the authors did their work? If it is the latter, then please delete the dates as it is confusing.

11.   Line 209: Is it 100,00 or 1000 inhabitants per day? Please revise throughout the manuscript and correct accordingly.

12.   Discussion (line 334): Please comment on the significant use of fluoroquinolones in this study. What is your assumption for this use? Do you think this rate of use was necessary or perhaps unnecessary and requires control? I also suggest discussing the current state of multidrug-resistant TB in your country. Could the extra use of moxifloxacin be attributed to that? Try to support your claims with references as possible.

13.   Discussion (lines 366-375): When discussing the overuse of antibiotics during COVID-19 pandemic, I suggest also discussing the overall global use and the subsequent impact on antimicrobial resistance. A good reference on that is: https://pubmed.ncbi.nlm.nih.gov/34473285/

Minor comments:

14.   Title: add the word "between" before the year range and remove the comma after 2021

15.   Abstract: Please use small letters for the names of the antibiotics.

16.   Line 60: Correct to "Keywords"

17.   Introduction: Combine line 66 to the previous paragraph.

18.   Introduction (line 105): Please spell out DID at its first occurrence. You can then use the abbreviation on line 131 without the spell out.

19.   Lines 147, 156, 160, and 200: Replace "is" with "are" after "data"

20.   Lines 153-154: Please rephrase this statement as it doesn't sound grammatically correct.

21.   Line 159: Just use the abbreviation since it was already spelled out previously.

22.   Line 210-211: Correct it to "overall antibiotic consumption"

23.   Line 248: Breakdown is a single word. Please correct.

24.   Line 334: I think you mean "increased"

25.   Line 373: There seem to be a "3" in the middle of the word "patients"

26.   Line 436: Add a full stop before "We"

27.   General: I suggest that the authors use "co-trimoxazole" instead of trimethoprim-sulfamethoxazole. They may include it in between parentheses after the first mentioning of the full name of the antibiotic on line 206.

Author Response

Reviewer 1

Edits

Comments

1.          Abstract: The abstract is too long as the word count exceeds the journal's recommendation of 200 words.

Please revise and shorten it. Also, remove the subheadings as it should be unstructured.

-          Reduced to 210.

-          Subheadings were removed.

2.          Introduction (line 83): Please cite the GLASS-AMC reference.

Included as references 3 to 6.

3.          Introduction (line 93): Please cite the WHO report.

done

4.          Methods (line 139): Please cite the website.

10 -12

5.          Table 1: Please cite the reference of this classification.

Cited as 9 for AWaRe

6.          Tables 2 and 3: I think it would be better to report the numbers in three decimals only instead of four (three numbers after the decimal point). This could also help shrink the table and allow space for other columns (i.e., active pharmaceutical ingredient). The same applies to the whole text of the manuscript.

done

7.          Part 3.1 (Training) is a duplicate. Please remove it from here. However, you may copy the procedure of the training to part 2.2 in the methods.

done

8.          Results (lines 200-207): This paragraph belongs to the methods section. It should be moved to section 2.3 while avoiding duplicate info.

removed

9.          Line 200: Please justify why the data were not collected from the same period 2017-2021? Or is this the period when the authors did their work? If it is the latter, then please delete the dates as it is confusing.

Deleted to avoid confusion. This is retrospective data collection.

10.      Line 209: Is it 100,00 or 1000 inhabitants per day? Please revise throughout the manuscript and correct accordingly.

1000 is correct.

This has been revised

11.      Discussion (line 334): Please comment on the significant use of fluoroquinolones in this study. What is your assumption for this use? Do you think this rate of use was necessary or perhaps unnecessary and requires control? I also suggest discussing the current state of multidrug-resistant TB in your country. Could the extra use of moxifloxacin be attributed to that? Try to support your claims with references as possible.

This could have been because moxifloxacin is reserved, at the ERHA, for hospital use as it is a broad-spectrum antibiotic. This would have helped in stewardship efforts to preserve its use for respiratory illness and complicated intra-abdominal infections, presenting to hospital.

We would like to stay away from TB as this is under a separate director and under separate control. We mention it diplomatically from publications.

12.      Discussion (lines 366-375): When discussing the overuse of antibiotics during COVID-19 pandemic, I suggest also discussing the overall global use and the subsequent impact on antimicrobial resistance. A good reference on that is: https://pubmed.ncbi.nlm.nih.gov/34473285/

done

Minor Comments Reviewer 1

13.      Title: add the word "between" before the year range and remove the comma after 2021

done

14.   Abstract: Please use small letters for the names of the antibiotics.

done

15.   Line 60: Correct to "Keywords"

done

16.   Introduction: Combine line 66 to the previous paragraph.

done

17.   Introduction (line 105): Please spell out DID at its first occurrence. You can then use the abbreviation on line 131 without the spell out.

done

18. Lines 147, 156, 160, and 200: Replace "is" with "are" after "data"

Changed to pleural

19.   Lines 153-154: Please rephrase this statement as it doesn't sound grammatically correct.

done

20.   Line 159: Just use the abbreviation since it was already spelled out previously.

done

21.   Line 210-211: Correct it to "overall antibiotic consumption"

done

22.   Line 248: Breakdown is a single word. Please correct.

done

23.   Line 334: I think you mean "increased"

Changed

24.   Line 373: There seem to be a "3" in the middle of the word "patients"

changed

25.   Line 436: Add a full stop before "We"

Conclusion changed as per instructions.

27.   General: I suggest that the authors use "co-trimoxazole" instead of trimethoprim-sulfamethoxazole. They may include it in between parentheses after the first mentioning of the full name of the antibiotic on line 206.

This was changed to co-trimoxazole

Reviewer 2 Report

Overall very well-conducted study which reflects well in the manuscript submitted. I have the following comments for your consideration.

Abstract

Line 41. "....as little as.." Not needed here as there is no comparator in this sentence. Removing it does not change the message as it is.

Line 53

Remove "use" after azithromycin since "consumption" at the beginning of the sentence conveys the same message.

Introduction

The numbering of the references needs to be checked. the first reference on line 65 is "27" and the next is "1" on line 67 then to "29" (line 87) to "9" (line 90), to "8" (line 95), and so on.

Kindly revise references throughout the manuscript.

Line 72. Insert "in" between trends and dosing or reword for clarity.

Lines 78-83

Provide a reference to WHO GLASS AMC or appropriate WHO citation

Line 89 sentence has to be in the past tense. eg: .....thought to be the most "challenged"......

Line 89 standardization of what? Clarify

Line 93

insert a reference for the report

Line 99 Sentence is wordy and can be split in two for clarity and ease of understanding for readers.

Methods. 

Since all activities have already been carried out, present in the past tense and not present. For example; in line 108, "This is"...should be "this was"

Lines 153 to 154 Kindly review the sentence for clarity. "Data was 'registered' in Ms Excel ......' tools' developed by the 'WHO AMC Template'...."

Section 2.3 Data

Provide additional information on the number and selection of community pharmacies. The map is not clear.

3.1 Training

training is singular in sentence two before going on to mention meetings (plural) and sessions. Confirm and revise.

Extensive text in the results section seems to be discussing the findings which would have been better suited to the discussion. Kindly consider with your discretion reducing the text if possible. For example, words like "interestingly" in line 230, can be removed. 

Line 243 Typo. I believe the authors meant "THE" and not "He"

Discussion

An initial paragraph summarising the key findings before describing them in detail in light of other studies would improve the discussion considerably although the current version is still appropriate. It gives readers with little time key points to take away as well as provides an order to the discussion as the findings would typically be listed in order of significance.

Words like interestingly, notably, etc could be removed so the authors go straight to the point. They do not change anything in their absence.

Line 411. Review sentence for clarity. Do the authors mean "A tool such as this ...." ?

Limitation

If study is not representative of the entire country's antibiotic consumption, should the authors be referring to this study earlier as the "first national " one conducted? Kindly review and revise.

Conclusion

This section needs to be straight to the point regarding the findings. As it stands it reads like a continuation of the discussion. For example, line 434; "This is particularly concerning due to the..." and line 436 "We also noted the decrease......" are not needed in the conclusion.

Author Response

Reviewer 2

Edits

Comments

Abstract

Line 41. "....as little as.." Not needed here as there is no comparator in this sentence. Removing it does not change the message as it is.

removed

Remove "use" after azithromycin since "consumption" at the beginning of the sentence conveys the same message.

done

Introduction

The numbering of the references needs to be checked. the first reference on line 65 is "27" and the next is "1" on line 67 then to "29" (line 87) to "9" (line 90), to "8" (line 95), and so on.

Kindly revise references throughout the manuscript.

revised

Line 72. Insert "in" between trends and dosing or reword for clarity.

done

Lines 78-83

Provide a reference to WHO GLASS AMC or appropriate WHO citation

Line 89 sentence has to be in the past tense. eg: .....thought to be the most "challenged"......

done

Line 89 standardization of what? Clarify

AMC data collection and presentation

Line 93

insert a reference for the report

done

Line 99 Sentence is wordy and can be split in two for clarity and ease of understanding for readers.

done

The data from this study therefore provided an avenue to assess antimicrobial consumption within the Caribbean setting. It formed the basis of inferences regarding the risk of multidrug resistance and inappropriate antimicrobial use. This data formed the basis of providing information for decision making such as policy prioritization.

Methods

Since all activities have already been carried out, present in the past tense and not present. For example; in line 108, "This is"...should be "this was"

done

Lines 153 to 154 Kindly review the sentence for clarity. "Data was 'registered' in Ms Excel ......' tools' developed by the 'WHO AMC Template'...."

done

Section 2.3 Data

Provide additional information on the number and selection of community pharmacies. The map is not clear.

This is done

Training

training is singular in sentence two before going on to mention meetings (plural) and sessions. Confirm and revise.

Line 243 Typo. I believe the authors meant "THE" and not "He"

done

Extensive text in the results section seems to be discussing the findings which would have been better suited to the discussion. Kindly consider with your discretion reducing the text if possible. For example, words like "interestingly" in line 230, can be removed.

Done as best as possible, without compromising content.

Discussion

An initial paragraph summarising the key findings before describing them in detail in light of other studies would improve the discussion considerably although the current version is still appropriate. It gives readers with little time key points to take away as well as provides an order to the discussion as the findings would typically be listed in order of significance.

Extensively revised

Words like interestingly, notably, etc could be removed so the authors go straight to the point. They do not change anything in their absence.

Line 411. Review sentence for clarity. Do the authors mean "A tool such as this ...." ?

Adjusted. We refer to the tools provided by CUFAR.

A tool such as this, provided by CUFAR, allows for a powerful level of quality assurance and standardization. It promoted a strengthened clinical governance in the Antimicrobial Resistance section of the Heath System.

Limitation

If study is not representative of the entire country's antibiotic consumption, should the authors be referring to this study earlier as the "first national " one conducted? Kindly review and revise.

Thanks for the comment.

No, but it is sentinel surveillance for the country. It is the country’s sentinel point and starting point for a National Initiative, we still need to capture that.

Conclusion

This section needs to be straight to the point regarding the findings. As it stands it reads like a continuation of the discussion. For example, line 434; "This is particularly concerning due to the..." and line 436 "We also noted the decrease......" are not needed in the conclusion.

done

Round 2

Reviewer 1 Report

I appreciate the edits made by the authors to improve their manuscript. I think it can be accepted in it's current format.